# Studies on the Relationship between Social Anxiety and Excessive Smartphone Use and on the Effects of Abstinence and Sensation Seeking on Excessive Smartphone Use

**DOI:** 10.3390/ijerph17041262

**Published:** 2020-02-15

**Authors:** Liat Turgeman, Inbar Hefner, Maayan Bazon, Or Yehoshua, Aviv Weinstein

**Affiliations:** Department of Behavioral Science, Ariel University, Science Park, Ariel 40700, Israel; liatshemesh1991@gmail.com (L.T.); ihifm88@gmail.com (I.H.); maayanbazon@gmail.com (M.B.); ory2507@gmail.com (O.Y.)

**Keywords:** smartphone use, social anxiety, abstinence, sensation seeking

## Abstract

Problematic smartphone use is the excessive use of the smartphone with negative impacts on the quality of life of the user. We investigated the association between social anxiety and excessive smartphone use. The sample consisted of 140 participants, 73 male and 67 female university students with a mean age of 26 years and 4 months (SD = 3.38), who filled in the Liebowitz Social Anxiety Scale and the Smartphone Addiction Scale (SAS). Results showed a positive association between social anxiety and excessive smartphone use. Social anxiety explained 31.5% of the variance of ratings on the SAS. A second study investigated the interaction between abstinence and sensation seeking and excessive smartphone use. The sample consisted of 60 participants, 44 female and 16 male university students. The sample was divided into two experimental conditions: 30 participants were abstinent for 1.5 h from the smartphone and 30 participants were non-abstinent. Results showed that excessive smartphone use increased in the group that abstained compared to those who did not. Secondly, participants who had high baseline sensation-seeking ratings had higher scores of excessive smartphone use after abstinence compared with those with low ratings of sensation seeking. These studies indicate the contribution of social anxiety to problematic smartphone use and how it can be exacerbated by the combination of abstinence and high sensation seeking.

## 1. Introduction

Behavioral addictions are characterized by excessive use despite adverse consequences, withdrawal phenomena, and tolerance that are also typical of substance use disorders. There is a debate about whether Internet and gaming disorder (IGD) should be classified as a behavioral addiction. In the fifth edition of the Diagnostic and Statistical Manual of Mental Disorders (DSM-5) [1], IGD is identified in Section III as a condition warranting further clinical research and experience before it might be considered for inclusion as a formal disorder. The inclusion of gaming disorder (GD) in the 11th revision of the International Classification of Diseases (ICD-11), developed by the World Health Organization (WHO) [2], has led to a lively debate in the last year. Although there has been broad support for the decision among academics, others have criticized this decision [3]. A large group of researchers have argued that including IGD will facilitate treatment and prevention for individuals with IGD [4].

We now witness a behavioral addiction to technologies such as television, video games, and the Internet [5,6,7]. Internet addiction is defined as “The inability to control one’s use of the Internet, a condition that causes severe impairment of various life functions” [8]. Internet-addicted individuals use the Internet for long periods of time, resulting in isolation from other social connections and a focus on Internet-related activities rather than engaging in real-life activities [7].

Smartphones are mobile phones which contain applications that enable the use of the Internet for activities such as gambling, social networking, shopping, etc. [9]. Using these applications, smartphones enable Internet-based communication [8]. Previous studies have investigated the frequency of smartphone use and resulting effects on mental and physical health measures [10,11,12,13,14].

Smartphones are computers that are advanced means of information technology, including access to the Internet and mobile communication [15]. The smartphone has become the main tool enhancing Internet addiction in all ages. Previous studies have researched Internet addiction by using smartphones, and they have concentrated on the student population [16].

There is controversy over using the term “smartphone addiction”, and Billieux et al. [9] have argued that there are no studies supporting the idea of mobile phone addiction. Smartphones are based on web-based applications (apps) and therefore they require skills for Internet use. Due to the likely relationship between the frequent use of smartphones and the Internet, excessive smartphone use was studied in the context of Internet addiction [8]. Recent research has shown a positive association between Internet addiction and excessive smartphone use, indicating that an excessive use of smartphone applications such as social media apps can predict an Internet addiction [5,17].

We have previously shown an association between Internet addiction and social anxiety, thus supporting previous evidence for their co-morbidity [7]. There is a current gap in the existing evidence for an association between excessive smartphone use and social anxiety. Based on our finding of an association between Internet addiction and social anxiety, this study investigated whether there is a positive association between problematic smartphone use and social anxiety. Secondly, little is known about the effects of abstinence from smartphone use and which factors contribute to excessive smartphone use. In order to fill this gap, this study investigated whether abstinence from smartphone use and sensation seeking have an effect on problematic smartphone use. We predicted that problematic smartphone use would be influenced by abstinence and by the level of sensation seeking of the participants.

## 2. Study 1

### 2.1. Materials and Methods

In this study, there were 140 participants: 73 male and 67 female university students from the Faculty of Social Science with a mean age of 26 years and 4 months (SD = 3.38) and an age range of 22 to 35 years old, who were recruited using convenience sampling. Socio-economic status was not available for analysis. 

### 2.2. Questionnaires

The study included a demographic questionnaire which asked for information about sex, age, year of birth, country of birth, marital status, employment, education, and use of the Internet.

#### 2.2.1. The Liebowitz Social Anxiety Scale

The Liebowitz Social Anxiety Scale (LSAS) [18] was used to measure anxiety and avoidance of social situations, and it was previously validated [18,19,20]. The questionnaire includes 24 items: 13 describe situations that require performance and 11 describe social interactions. The responses of social fear or social avoidance range from 1 (never) to 4 (very much). In our study, results had a Cronbach internal consistency of α = 0.971, social fear α = 0.951, and social avoidance α = 0.935.

#### 2.2.2. The Smartphone Addiction Scale

The Smartphone Addiction Scale (SAS) [17] measures problematic smartphone use. The SAS includes 33 statements. Answers are on a Likert scale between 1 and 6, where 1 is “strongly disagree” and 6 is “strongly agree”. The sum of responses was divided to three levels: low addiction (1–66), medium addiction (67–132), and high addiction (133–198). In this sample, there was a Cronbach internal consistency of α = 0.96. The SAS was first introduced, validated, and correlated with a visual analog scale (VAS) questionnaire (Kwon et al., 2013) [17]. The questionnaires were delivered on the Internet.

### 2.3. Statistical and Data Analysis

The analysis of the results was performed in SPSS (IBM Corp., Armonk, NY, USA). Correlation analysis was performed on questionnaire measures for all participants. A multiple regression analysis was performed to assess the contribution of social anxiety to the variance of problematic smartphone use and the contribution of problematic smartphone use to the variance of social anxiety.

### 2.4. Ethics

The study was approved by the Institutional Review Board (IRB, Helsinki committee) of Ariel University AU-AW-20180312 and AU-SOC-AW-20190318. All participants signed an informed consent form.

### 2.5. Procedure

Participants completed the LSAS [18] and SAS [17].

### 2.6. Results

Questionnaire ratings in all participants are shown in Table 1.

Table 2 shows correlations between all questionnaires in all participants.

### 2.7. Treatment of the Hypotheses

There was a positively high Pearson correlation (*r* = 0.562; *p* < 0.01) between problematic smartphone use and social anxiety, suggesting that problematic smartphone use is associated with high levels of social anxiety. There was a positive moderate Pearson correlation (*r* = 0.569; *p* < 0.01) between problematic smartphone use and the social fear subscale of the LSAS and a positive moderate Pearson correlation (*r* = 0.537; *p* < 0.01) between problematic smartphone use and the social avoidance subscale of the LSAS, both suggesting that problematic smartphone use is associated with moderate levels of social anxiety. See Figure 1 for the correlation between problematic smartphone use and social anxiety.

A multiple regression analysis showed that a model which included social anxiety (*β* = 0.562, R^2^_adj_ = 0.31, *p* < 0.001) scores contributed significantly to the variance of problematic smartphone use ratings (F(1,138) = 63.56, *p* < 0.001) and explained 31.5% of the variance of these ratings. Problematic smartphone use predicted social fear (*β* = 0.569, R^2^ = 0.323, *p* < 0.001) and explained 32.3% of the variance of social fear. Additionally, problematic smartphone use predicted social avoidance (*β* = 0.537, R^2^ = 0.288, *p* < 0.001) and explained 28.8% of the variance of social avoidance.

## 3. Study 2

### 3.1. Materials and Methods

The sample consisted of 60 participants, 44 female and 16 male university students with a mean age of 23 years and 10 months (SD = 2.22) and an age range of 19 to 30 years old, who were recruited by convenience sampling among friends and fellow students. Participation was rewarded with coupons distributed to students in experiments at the university.

### 3.2. Questionnaires

#### 3.2.1. Smartphone Addiction Scale (SAS)

The SAS [17] measured problematic smartphone use described in the first study. This questionnaire had a Cronbach internal consistency of α = 0.94 in this study.

The smartphone addiction (VAS short) questionnaire [17] consisted of seven questions on problematic smartphone use. The questionnaire measured the subjective thoughts of the participants on their smartphone use. Each item was scored between 1 and 11 points, varying from 1 (not at all) to 11 (very much). Ratings were divided between low addiction (1–26), moderate addiction (27–52), and high addiction (53–77). This questionnaire had a Cronbach internal consistency of α = 0.91 in this study.

#### 3.2.2. Sensation Seeking Scale

The Sensation Seeking Scale (SSS) by Zuckerman [21] includes 40 items where participants had to choose between two opposite items. There were four personality traits assessed: disinhibition, boredom susceptibility, thrill and adventure seeking, and experience seeking. The questionnaire was validated previously [22] and it showed Cronbach’s internal consistency of α = 0.83–0.86. In our study, this questionnaire had a Cronbach internal consistency of α = 0.81.

### 3.3. Statistical and Data Analysis

The analysis of the results was performed on SPSS (IBM Corp., Armonk, NY, USA). A two-way repeated measures analysis of variance (ANOVA) was used to measure the interaction between abstinence and sensation seeking on ratings of the smartphone addiction (VAS short) questionnaire. A follow-up analysis compared the effects of abstinence and sensation seeking on ratings of smartphone addiction and it compared scores on smartphone addiction in abstinent and non-abstinent participants.

### 3.4. Ethics

The study was approved by the Institutional Review Board (IRB, Helsinki committee) of Ariel University AU-SOC-AW-20190318. All participants signed an informed consent form.

### 3.5. Procedure

Participants completed the smartphone addiction (VAS short) questionnaire, SAS [17], and SSS [21] before the experiment. The sample was divided into two experimental conditions: 30 participants were abstinent for 1.5 h from the smartphone and 30 participants were a non-abstinent control group. After 1.5 h, the participants filled in the smartphone addiction (VAS short) questionnaire.

### 3.6. Results

Table 3 below shows descriptive characteristics of the key study variables.

The participants showed a moderate level 39.60 (SD = 10.83) of smartphone addiction according to the smartphone addiction (VAS short) questionnaire. The addiction level of the participants to smartphones on the SAS was moderate, with an average of 95.7 (SD = 25.89) and the range of scores between 38 and 196.

### 3.7. Treatment of the Hypotheses

The first hypothesis predicted that abstinence and sensation seeking would affect ratings on the smartphone addiction (VAS short) questionnaire. A two-way repeated measures ANOVA showed an interaction between abstinence and sensation seeking on ratings of the smartphone addiction (VAS short) questionnaire F(1,18) = 2.698, *p* < 0.05. A follow-up comparison showed that abstinence had a significant effect on ratings of the smartphone addiction (VAS short) questionnaire F(1,18) = 58.590, *p* < 0.01, but sensation seeking had no significant effect on ratings of the smartphone addiction (VAS short) questionnaire F(1,18) = 1.131, *p* > 0.05. Furthermore, a simple comparison showed that participants after abstinence had higher scores on the smartphone addiction (VAS short) questionnaire than the non-abstinent control group t(1,58) = 2.109, *p* < 0.05.

Figure 2 shows ratings on the smartphone addiction (VAS short) questionnaire before and after abstinence in the abstinent group and the non-abstinent control group.

The second hypothesis predicted differences in ratings on the smartphone addiction (VAS short) questionnaire depending on levels of sensation seeking. Ratings on the smartphone addiction (VAS short) questionnaire were predicted to be highest in participants who scored high on the sensation seeking questionnaire.

A dependent sample *t*-test examined the differences in ratings on the smartphone addiction (VAS short) questionnaire in high and low sensation seeking individuals before and after abstinence. Results showed that, at baseline, there was no difference between participants with high scores of sensation seeking and those with low scores of sensation seeking on the smartphone addiction (VAS short) questionnaire t(1,28) = 1.45, *p* = 0.158. After abstinence, participants with high scores of sensation seeking scored higher on the smartphone addiction (VAS short) questionnaire than those with low sensation seeking scores t(1,28) = 2.282, *p* < 0.05. See Figure 3 for scores on the smartphone addiction (VAS short) questionnaire, comparing high and low sensation seeking individuals after abstinence from smartphone use.

## 4. Discussion

The results of this study show for the first time that excessive smartphone use was correlated with high ratings of social anxiety, thus supporting our early results showing an association between Internet addiction and social anxiety [7]. According to this evidence, similar to previous evidence on Internet addiction, highly anxious individuals could prefer social communication that enables them to avoid and control negative aspects of their behavior or appearance. Hence, they may use smartphones excessively in order to escape from social situations; this behavior has major implications for social communication in young adults [23].

These conclusions are supported by previous research findings showing an association between poor social skills and excessive Internet use [24]. Additional research demonstrated that fear, anxiety, and depression were positively correlated with cognition about problematic Internet use in males [25]. Adolescents with Internet addiction have also shown high levels of social anxiety [26]. A positive correlation was shown between problematic Internet use and shyness, loneliness, avoidance of social relationships [27,28,29,30,31], and dating anxiety [32]. Problematic Internet users were more neurotic, less extraverted, more socially anxious, lonelier, and gained greater support from social networks on the Internet than non-problematic Internet users [33].

The correlation between social anxiety and excessive smartphone use was moderate. This indicates that individuals with social anxiety may feel ashamed of their functional and emotional problems in social situations. Social anxiety is associated with a lack of confidence in presentation skills and an effort to create a positive impression on others. In order to reduce anxiety, socially anxious individuals can prefer encounters with low-risk social communication. Online interaction is preferred to face-to-face communication since it involves a lower risk and it enables socially anxious individuals to hide and control the less-positive aspects of their appearance and behavior [34]. Furthermore, lonely or socially anxious students can develop compulsive Internet use behaviors resulting in harm to other significant activities such as work, school, or significant relationships, instead of solving their original problems. These negative outcomes isolate socially anxious individuals from healthy social activities and may result in loneliness [35]. These studies support our finding that high levels of social anxiety were associated with excessive smartphone use.

There are several explanations for the association between excessive Internet use and social anxiety. The social compensation hypothesis suggests that the Internet benefits individuals who feel uncomfortable in face-to-face communication. Students with high social anxiety in the Midwest (USA) have shown greater feelings of comfort and self-disclosure when socializing online and less self-disclosure when communicating face-to-face, compared to less socially anxious individuals [36]. Social anxiety was associated with lower quality of life, higher rates of depression, and poorer well-being in individuals who communicated frequently online.

Socially anxious individuals find it easier to communicate online anonymously rather than engaging in face-to-face interaction due to fear of negative evaluation by others [37]. This relationship can be both beneficial and harmful, since those who suffer from social anxiety turn to the smartphone because it is easier for them to freely communicate, but excessive smartphone use can exacerbate social isolation in those who are already socially isolated. Finally, there is contrary evidence that using the Internet for social purposes can actually reduce social anxiety. Chatting online can be used for training and improving social skills [38].

Although social anxiety contributed to the variance of ratings of excessive smartphone use, it only accounted for 31.5% of the variance. Other factors such as depression, loneliness, anxiety, low sleep quality [12,39], addiction to social networking, shyness, and low self-esteem [40] may also contribute to the variance of excessive smartphone use.

Sensation seeking is the pursuit of new experiences and risk-taking in order to fulfill these experiences. This occurs through physical activity, exceptional lifestyle practices, and addiction to high-risk behavior. It is well established that sensation seeking is associated with addiction to drugs and to behavioral addictions such as gambling and dangerous sports. In our study, an interaction between abstinence, sensation seeking, and problematic smartphone use was found, which implies that the combination of high sensation seeking and abstinence increases excessive smartphone use. This interaction may be explained by boredom, avoidance of uncomfortable situations, and the need for entertainment [40]. The easy access and availability of the Internet enables users to send messages, play games, download information, and interact on social networkss; when these are temporarily unavailable, it may exacerbate the experience of boredom and the need for entertainment [41]. There is supporting evidence that young individuals with excessive smartphone use show high boredom ratings, have high ratings of sensation seeking, and use a wide range of smartphone applications [42]. An important ingredient in sensation seeking is susceptibility to boredom [10], and that may explain the interaction with abstinence that was found in our study. In our previous study [43], we found that excessive smartphone use was not influenced by any interest or involvement in daily activities during a lecture (active—drawing with a pen/pencil of a picture, passive—watching a film or a state of boredom, i.e., with no verbal communication with other participants). The results of this study indicate for the first time that a personality disposition for sensation seeking interacts with abstinence in order to exacerbate excessive smartphone use. The results of this study can therefore explain why some individuals become more addicted to smartphones than others.

### Limitations

The studies tested only university students, and that may not reflect the population as a whole. Secondly, the second study included only 60 participants and that may be too small a sample to draw strong conclusions. The questionnaires rely on self-rating, and are highly subjective. The abstinence period of an hour and a half was chosen, as it is the length of a university lecture. It seems to be long enough, but further studies may investigate longer periods of abstinence. Finally, the population of this study had a moderate use of smartphones. Future studies can examine young populations with varying levels of smartphone use.

## 5. Conclusions

The findings of this study suggest that the excessive use of smartphones requires attention like all behavioral addictions. Individuals with excessive smartphone use may have social and emotional problems and difficulties in adapting to and coping in daily life. They usually have low confidence and self-esteem and they struggle with social relationships [7,44]. Smartphones provide a means for escaping from social encounters in daily life into a virtual social world. Excessive smartphone use may be exacerbated in those with high sensation seeking following abstinence from smartphone use, due to boredom and the need for social stimulation. Excessive or problematic use of smartphones may impair social skills and the ability to understand and convey emotional and social information.

## Figures and Tables

**Figure 1 ijerph-17-01262-f001:**
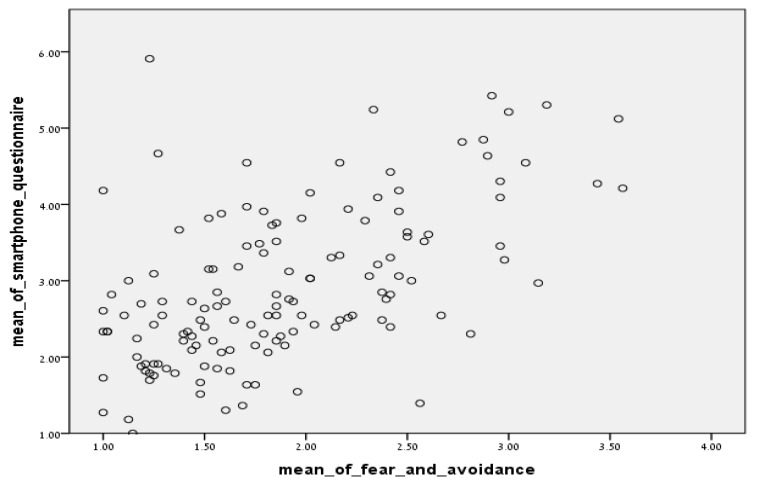
The association between problematic smartphone use and social anxiety.

**Figure 2 ijerph-17-01262-f002:**
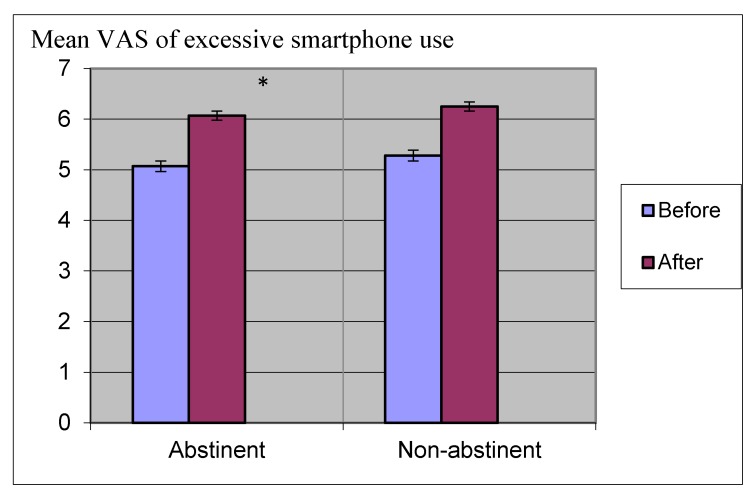
Ratings of the smartphone addiction (VAS short) questionnaire before and after abstinence in the abstinent group compared with the non-abstinent control group. * *p* < 0.05.

**Figure 3 ijerph-17-01262-f003:**
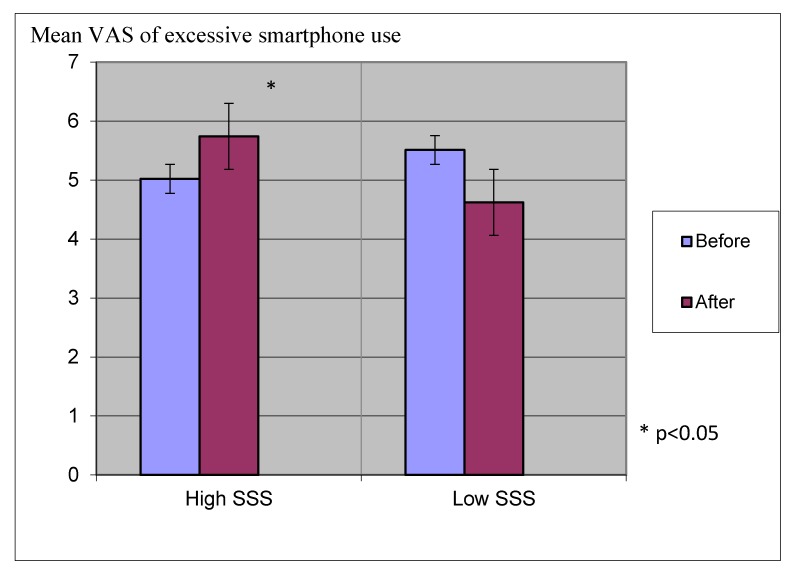
Scores of the smartphone addiction (VAS short) questionnaire, comparing high and low sensation seeking individuals after abstinence from smartphone use. SSS = Sensation Seeking Scale.

**Table 1 ijerph-17-01262-t001:** Study 1—Questionnaire ratings in all participants mean (SD).

Factor	Males (*n* = 73)	Females (*n* = 67)	Total (*n* = 140)
SAS	95.89 (36.88)	96.58 (29.79)	96.22 (33.56)
LSAS	1.85 (0.69)	1.91 (0.51)	1.88 (0.6)
LSAS—social fear	1.84 (0.73)	1.92 (0.53)	1.88 (0.63)
LSAS—social avoidance	1.87 (0.67)	1.90 (0.52)	1.84 (0.6)

SAS: Smartphone Addiction Scale (Kwon et al., 2013) [17]; LSAS: Liebowitz Social Anxiety Scale (Heimberg et al., 1999) [18]; LSAS—social fear: a subscale of the Liebowitz Social Anxiety Scale (Heimberg et al., 1999) [18] LSAS—social avoidance: a subscale of the Liebowitz Social Anxiety Scale (Heimberg et al., 1999) [18].

**Table 2 ijerph-17-01262-t002:** Study 1—Pearson *r* correlations between questionnaires and Smartphone Addiction Scale scores (*n* = 140).

Factor	SAS	LSAS	LSAS—Social Fear	LSAS—Social Avoidance
SAS				
LSAS	0.56 **			
LSAS—social fear	0.57 **	0.98 **		
LSAS—social avoidance	0.54 **	0.99 **	0.94 **	

SAS: Smartphone Addiction Scale (Kwon et al., 2013) [17]; LSAS: Liebowitz Social Anxiety Scale (Heimberg et al., 1999) [18]; LSAS—social fear: subscale of the Liebowitz Social Anxiety Scale (Heimberg et al., 1999) [18]; LSAS—social avoidance: subscale of the Liebowitz Social Anxiety Scale (Heimberg et al., 1999) [18]; ** *p* < 0.01.

**Table 3 ijerph-17-01262-t003:** Study 2—Questionnaire ratings in all participants *n* = 60 mean (SD).

Factor	All Subjects	Range
SAS	95.7 (25.89)	38–196
VAS addiction	39.6 (10.83)	21–69
SSS	20.2 (6.48)	5–38

SAS (Kwon et al., 2013) [17]; VAS: visual analog scale for smartphone addiction (Kwon et al., 2013) [17]; SSS: Sensation Seeking Scale (Zuckerman, 1964) [21].

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
