# Peer review of "Studies on the Relationship between Social Anxiety and Excessive Smartphone Use and on the Effects of Abstinence and Sensation Seeking on Excessive Smartphone Use"

_ijerph, 2020, doi:10.3390/ijerph17041262_

Round 1

Reviewer 1 Report

The authors have revised their manuscript based on the suggestions of reviewers.

Author Response

We thank the reviewer for the positive review

Reviewer 2 Report

I would like to thank the authors for their work, however I did not feel the authors made any significant improvements with regards to the main issues I raised in the first review. The problems remain the same. I think the manuscript must describe a technically sound piece of scientific research with data that supports the conclusions. Experiments must have been conducted rigorously, with appropriate controls, replication, and sample sizes. The conclusions must be drawn appropriately based on the data presented.

I regret that the disposition is not favorable, but would like to thank you for your support.

Merry christmas and prosperous 2020

Author Response

We regret that the reviewer has not found the scientific work adequate. We have done our best to respond to the comments raised in the first round.

In study 1- We have followed the scientific methods that we have described in our previous published study. We even added a more rigorous analysis which is regression analysis

Weinstein, A., Dorani, D., Elhadif, R., Bukovza Y, Yarmulnik, A., Dannon, P. Internet addiction is associated with social anxiety in young adults. (2015). Annals of Clinical Psychiatry, 27(1):4-9.

In the second study we have followed the scientific methods that we have described in our previous published study:

Ben-Yehuda, L., Greenberg, L., Weinstein, A. (2016). Internet addiction by using the Smartphone- relationships between Internet addiction, frequency of Smartphone use and the state of mind of male and female students. Journal of Reward Deficiency Syndrome and Addiction Science, 2 (1), 22-27.

Both are highly respected scientific journals in the fields of science and Psychiatry. We respectfully disagree with the reviewer's evaluation that the scientific method is not rigorous enough.

Merry Christmas and prosperous 2020

Reviewer 3 Report

This is a highly significant paper on a subject we don't hear nearly enough about.

But from my perspective as a reader who is outside his area of expertise, I felt that the conclusions were clear and important and that the experiments seemed to be well designed.

Author Response

We thank the reviewer for a positive review

Reviewer 4 Report

Thank you very much for the opportunity to review the article entitled "A study on the relationships between social anxiety, abstinence, sensation seeking and excessive smartphone use". 

I think the paper contributes to the existing literature, although it needs some modifications. However, if the authors would like to review some points, I think it is possible to publish in this journal. 

The assumptions of the manuscript are not very clear. Authors are invited to highlight the gaps in the literature and to specify the hypotheses of the work more clearly, highlighting the literature supporting the hypotheses. 

Moreover, the division into paragraphs is not consistent with the division between the two studies. Authors are invited to separate the methodological parts of the two studies (also Data analysis paragraph).

In both studies further information on the sample would be needed (which faculties are they enrolled in? Are they economically independent? Socio-economic status etc). In addition, demographic variables should be controlled in the analyses.

As regard the first study, I see that the results of a multiple regression are reported, but in the paragraph Data Analysis, it was not described.

Overall, the importance of greater consistency between the assumptions, the description of the methodology and the results is stressed. Likewise, in the discussions, more emphasis is placed on the innovative results of the work, comparing them with the literature.

Author Response

We are grateful for the reviewer's comments

We now highlighted the gaps in the current knowledge, hypothesses and supporting literature in the introduction

Secondly, we have now separated study 1 and study 2 methodologically and in terms of data analysis

Students were recruited from the Social science faculty at ariel university. They are all independent financially although they recieve support from their families. Current socio-economical status is not availabale for analysis. We have highlighted it in text.

We have added a description of correlation and regression analysis to the data analysis of study 1. We have also added a description of the statistics in study 2.

We have now emphasized the innovative aspect of the results in the discussion

Round 2

Reviewer 4 Report

I would like to thank the authors for the revisions they have made, which I think will allow greater transmission of the results that have emerged.

Author Response

Dear reviewer

We are greatful for your comments.

This manuscript is a resubmission of an earlier submission. The following is a list of the peer review reports and author responses from that submission.

Round 1

Reviewer 1 Report

This study contained two studies in smartphone use among university students. The first study investigated the association between social anxiety and excessive smartphone use. The second study investigated the interaction between abstinence and sensation seeking and excessive smartphone use.

This study had several major flaws.

These two studies had no connection with each other. They should be developed into two full original studies individually. These two studies were not fully introduced and described. There was no clear study hypothesis and frame. The measures (for example, the Smartphone addiction (VAS short) in study 2) and procedures were not clearly introduced. The contents of Introduction section did not well describe the results of previous studies on similar topics to serve as the basis of conducting this study, especially regarding to smartphone use abstinence and sensation seeking. The statistical strategies also needed further introduction.

Reviewer 2 Report

This study has attempted to examine the on the relationships between intimacy, sensation seeking and social anxiety and problematic smartphone use.
Although this is an novel observational study, I am struggling to understand the usefulness of this work. This might be due to the way the manuscript is presented.
I suggest that the authors consider the following:
· Provide a clear review of literature and provide a succinct overview of the work in this area related to intimacy, sensation seeking and social anxiety and problematic smartphone use,
· Indicate the gap in knowledge.
· Why do you think measuring to problematic smartphone use is important? Provide a robust argument. This will set the scene for your paper.
· Then outline your research question.
· In terms of methodology, this section is poor, needs to present a better rationale for the study and the methodology employed. Please, clearly explain what you have done. Include description of this protocol, its reliability and validity and the actual measurements.
Also, neither appear information related with inclusion and exclusion criteria, dates, protocol. The study design is a observational study of no ramdom sampling method, where the study was conducted in the hospital or in the outpatient center?.
Also, Please, expand and clarification information related with the calculate sample size.
Please include the date and code register number of ethics committee
· I am struggling to make sense of some of this, I am afraid it needs extensive revision . How did you calculate the results related with this protocol?
Please, provide clear results and describe them. Use appropriate statistics.
· Within your discussion, outline your results, discuss their novelty and their application to practice.
CONCLUSIONS: please include this section and reflect the study findings.
I regret that the disposition is not favorable, but would like to thank you for your support.
I wish you all the best.